# Large Language Models as Topological Thinkers: A Benchmark on Graph Persistent Homology

Hao Li [1]   Hao Wan [1]   Yixue Huang [1]   Yuzhou Chen [2]   Yulia R. Gel [3]   Hao Jiang [1]

## Abstract

Large language models (LLMs) are increasingly used in scientific discovery, system modeling, and decision-making, prompting interest in their ability to reason over complex structured data. Existing benchmarks primarily focus on static or local graph reasoning, overlooking the high-order structures in real-world systems whose global properties evolve across multiple scales. We introduce LLM4PH, a benchmark that evaluates multi-scale structural reasoning through the lens of persistent homology (PH), a topological framework for tracking structural evolution. LLM4PH decomposes the PH pipeline into interpretable reasoning tasks spanning synthetic and real-world graphs, revealing that most models struggle with reasoning over structural transitions and persistence. Beyond task-level evaluation, we perform cross-task ablations on prompt encoding and transfer, explore post-training effects, and construct a compositional PH pipeline to assess end-to-end performance. Our results provide the first in-depth view of how well LLMs bridge discrete graph structures with continuous topological abstraction, and offer insights into their potential for structure-aware scientific reasoning.

## 1. Introduction

Large Language Models (LLMs) have demonstrated remarkable performance across a wide range of natural language understanding and reasoning tasks (Chang et al., 2024; Hadi et al., 2023; Plaat et al., 2025). As LLMs are increasingly deployed in scientific analysis, complex system modeling, and agent-based decision making (Zheng et al., 2025; Besta et al., 2024; Lu et al., 2023), understanding their ability to reason over structured data has become a key focus of recent research.

Recent benchmarks have evaluated LLMs on graph-structured inputs by recasting classical problems, such as shortest path computation, connectivity, and traversal, into natural language tasks (Fatemi et al., 2024; Wang et al., 2023; Zhang et al., 2024). However, many real-world systems exhibit high-order structural patterns that evolve in coordination, such as communities in social networks, functional modules in biological interaction networks, and criminal gang formation (Dabaghian et al., 2012; Curto & Sanderson, 2025; Gravel & Bouchard, 2025). Accurately reasoning about these dynamic, multi-entity relationships is essential for understanding complex systems. While recent efforts have begun to address high-order structures using graph patterns and hypergraphs (Dai et al., 2025; Feng et al., 2025), such representations often lack a coherent notion of scale or temporal continuity, making it difficult to track how structure changes over time. In these settings, the core challenge shifts from answering static queries on a fixed graph to understanding how global structure emerges and evolves as the underlying system is progressively transformed.

We argue that persistent homology (PH) provides a principled and minimal abstraction for evaluating this missing capability. PH formalizes structure through a filtration process, where a system is gradually transformed and topological features such as connected components or cycles are tracked across the entire evolution (Edelsbrunner & Harer, 2010; Zomorodian & Carlsson, 2005; Carlsson, 2009). Recent studies show that PH can reveal subtle graph patterns overlooked by conventional methods, including structural signals in blockchain transaction graphs related to illegal and illicit activity (Akcora et al., 2020; Ofori-Boateng et al., 2021; Fan et al., 2022; Nakatani et al., 2024; Yue et al., 2024; Chen et al., 2025; Dey & Gel, 2026). These applications also expose a central design challenge. The behavior of PH depends on choices such as the filtration function, filtration schedule, and graph to complex construction. A useful model must therefore connect local graph evidence with a global topological objective and track how the induced structure changes across scales. Rather than treating PH only as a specialized mathematical tool, we use it as a lens

[1] Wuhan University, Wuhan, China [2] University of California, Riverside, USA [3] Virginia Tech, Blacksburg, USA. Correspondence to: Hao Jiang <jh@whu.edu.cn>.

*Proceedings of the 43rd International Conference on Machine Learning*, Seoul, South Korea. PMLR 306, 2026. Copyright 2026 by the author(s).

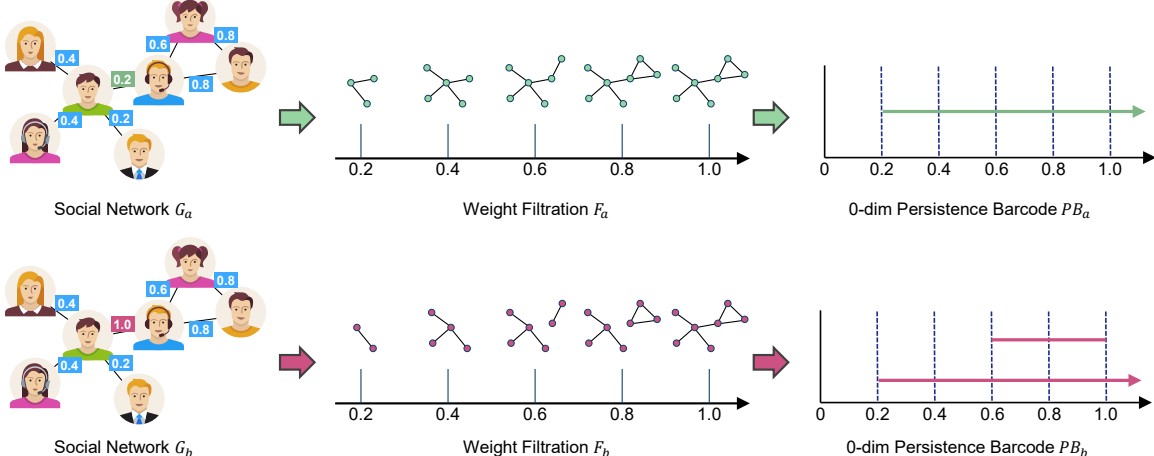

*Figure 1.* A toy example illustrating persistent homology as a lens for LLM graph reasoning. Two weighted graphs with similar local structure can induce different filtration trajectories and 0-dimensional persistence barcodes. This requires the model to track structural evolution rather than answer only a static graph query.

for probing whether LLMs can reason about structured state transitions and global constraints. Additional background on PH is provided in Appendix C.

Figure 1 illustrates this distinction with a small weighted social graph example. Although the two graphs share similar local structure, their filtrations evolve differently as edges are added by threshold. One graph remains connected throughout the process, whereas the other contains components that merge only later. The resulting 0-dimensional persistence barcodes are therefore different. This example shows that PH probes structural evolution across scales, which requires a model to track state changes rather than answer only a fixed graph query.

Building on this perspective, we introduce **LLM4PH**, a benchmark for evaluating whether LLMs can reason about multi scale structural evolution in graph persistent homology. To our knowledge, it is the first benchmark that treats PH on graphs as a structured reasoning problem for LLMs rather than only as a downstream computational tool. LLM4PH decomposes the PH workflow into interpretable sub-tasks, from simplicial structure recognition to filtration evolution, filtration strategy design, and persistence based graph inference. It further includes terminology ablation, compositional pipelines, and real-world graph settings to examine whether LLMs understand topological principles or rely on shallow pattern matching. Our results show that current models can handle some local structures and strongly scaffolded prompts, but still struggle with structural evolution and scale interactions.

## 2. Benchmark Design

As shown in Figure 2, LLM4PH is organized around the computational stages of graph persistent homology. The benchmark starts from simplicial structure recognition, moves to filtration evolution reasoning, and then evaluates filtration strategy design and real world graph inference. This progression separates local graph understanding from the more difficult ability to maintain a topological state over an evolving complex.

The benchmark is constructed to make each evaluated reasoning target explicit and reproducible. Ground truth labels are generated algorithmically from PH computations rather than human or LLM annotation. Synthetic graphs are sampled with balanced size groups and then filtered to retain instances with verifiable topological events, nontrivial filtration behavior, and well defined answers. Real world instances are selected from public graph classification datasets with clean labels and graph sizes that can be represented in text. Each graph is converted into controlled prompt formats so that graph representation and task formulation can be studied as evaluation variables rather than hidden confounds. Additional construction details are provided in Appendix B.

Task difficulty is defined by the stage of the PH workflow and the amount of structured state required from the model. Simple tasks focus on static recognition before filtration. Medium tasks require tracking births, merges, and component counts across a filtration sequence. Hard tasks require reasoning backward from a topological objective to a filtration strategy, which introduces a larger search space and a more abstract decision target.

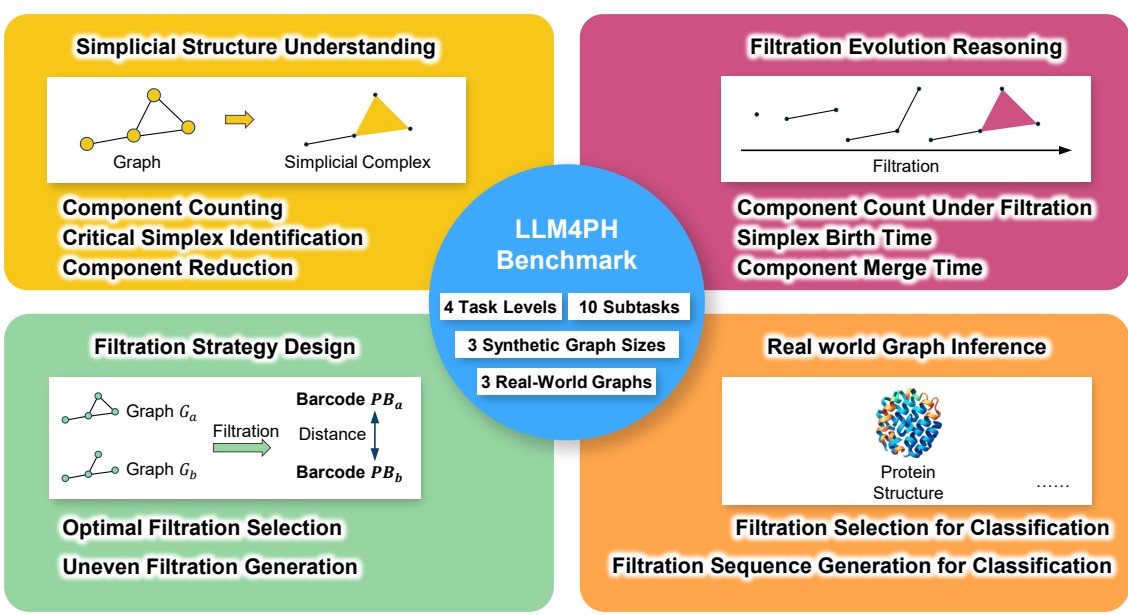

*Figure 2.* Benchmark Design.

## 2.1. Task Design

To assess the capability of LLMs in graph PH reasoning, we define four progressively challenging task categories. All tasks use the Vietoris–Rips (VR) complex as the graph induced simplicial complex, following common practice in graph persistent homology. Further details on task construction are provided in Appendix D.

**Simple Tasks (Simplicial Structure Understanding).** These tasks examine whether LLMs can recognize topological features such as connected components and critical simplices before the filtration. Tasks are evaluated by accuracy.

From the perspective of PH, these tasks correspond to the input stage before filtration begins. The model must identify the fundamental simplicial elements, such as 0-simplices and 1-simplices, that later form the basis of the topological space. Accurate understanding at this level is necessary before any meaningful filtration reasoning can occur.

- *0D Component Counting.* Counts the number of connected components (0-dimensional homology classes).
- *1D Simplex Counting.* Counts the number of 1-simplices (edges).
- *Component Reduction.* Modifies one edge in a given graph to reduce the number of connected components.

**Medium Tasks (Filtration Evolution Reasoning).** These tasks assess whether LLMs can reason about births and deaths of topological features. Tasks are evaluated by accuracy.

In the context of PH, these tasks correspond to tracking the dynamic evolution of topological features as the filtration progresses. The goal is to test whether LLMs can associate filtration steps with topological events such as component merging or cycle formation, which are essential for constructing persistence diagrams.

- *Simplex Birth Time.* Given a graph and its filtration sequence, predicts the birth time of specific simplices.
- *Component Merge Time.* Given a graph and its filtration sequence, predicts when specific connected components merge.
- *Component Count Under Filtration.* Given a graph and its filtration sequence, determines the number of connected components at a specified filtration value.

**Hard Tasks (Filtration Strategy Design).** These tasks challenge LLMs to select or generate filtration strategies that maximize the Wasserstein distance between the persistence barcodes of two given graphs. We consider both global filtration function selection and non-uniform threshold generation under a fixed filtration function. Tasks are evaluated using ranking-based metrics. From the PH perspective, this stage focuses on filtration design, which directly determines how simplicial complexes evolve and which topological features are revealed. LLMs are expected to reason backward from topological goals, such as maximizing feature differences, to suitable filtration strategies.

- *Optimal Filtration Selection.* Given two distinct graphs, the model selects the optimal filtration function from node degree, edge weight, K-shell, closeness centrality, be-

tweenness centrality, and eigenvector centrality to maximize the Wasserstein distance between their resulting persistence barcodes.

- *Non-Uniform Filtration Generation.* Given two distinct graphs with edge weights ranging from 1 to 10, and a fixed filtration function (edge weight), the model generates a non-uniform filtration sequence of length 5 (e.g., (1, 3, 4, 7, 10)) that maximizes the Wasserstein distance between their persistence barcodes.

**Real-World Tasks (Real-World Graph Inference).** These tasks evaluate the capability of LLMs to transfer and apply PH insights acquired from synthetic scenarios to real-world graph datasets. Specifically, we test whether LLMs can choose or generate suitable filtration strategies to solve real-world graph classification tasks. Tasks are evaluated by classification accuracy. The datasets are from (Sutherland et al., 2003), corresponding to molecular graphs targeting benzodiazepine receptor (BZR), cyclooxygenase-2 (COX2), and dihydrofolate reductase (LDHFR), respectively.

This stage reflects the final step of the PH pipeline, where topological features are used for downstream tasks such as classification. It tests whether LLMs can operationalize abstract persistence information in practical settings.

- *Filtration Selection for Classification.* Given four graphs from two classes, the model selects the optimal filtration function from node degree, edge weight, K-shell, closeness centrality, betweenness centrality, or eigenvector centrality to group the graphs correctly by class.
- *Filtration Sequence Generation for Classification.* Given four graphs from two classes and a fixed filtration function with a set of candidate filtration values, the model generates a filtration sequence that enables correct classification of the four graphs into two groups.

## 3. Experiments

Based on the LLM4PH benchmark, we conduct a series of experiments to evaluate the performance of various large language models across different tasks. The goal is to assess whether LLMs are capable of understanding the abstract concepts behind PH and applying this knowledge to practical graph analysis tasks.

**Experimental Setup.** We evaluate nine LLMs including GPT-4.1-mini (gpt-4.1-mini-2025-04-14), GPT-4o (gpt-4o-2024-08-06), Claude (claude-3-7-sonnet-20250219), Gemini-2.5 (gemini-2.5-flash-preview-04-17), DeepSeek-R1, DeepSeek-V3, Qwen3-30B, Mistral-small-3.1-24B, and Llama-3-70B. GPT, Claude, Gemini, and DeepSeek models are evaluated through hosted inference services. Qwen, Mistral, and Llama are deployed locally on a Linux server equipped with four NVIDIA A6000 GPUs. We additionally evaluate GraphWiz (Chen et al., 2024a) as a graph instruction following baseline, with details and results reported in Appendix F.1. In some experiments, `Random` denotes predictions sampled uniformly from the empirical label distribution.

### 3.1. Simple Tasks: Simplicial Structure Understanding

**Performance analysis.** Simple tasks in our benchmark are conceptually related to prior LLM benchmarks on graphs that focus on counting connected components or identifying cycles (Fatemi et al., 2024; Wang et al., 2023). However, instead of relying purely on graph-theoretic terminology, our benchmark frames these problems within the language of PH. For example, connected components are treated as 0-dimensional homology classes, and edges are seen as 1-simplices forming the foundation of higher-order topological features. This reframing not only allows for a more unified multi-scale perspective, but also tests whether LLMs can map familiar combinatorial structures into a topological reasoning framework.

As shown in Table 2, most LLMs perform very well on the *0D Component Counting* task. Deepseek-V3 and Gemini reach near-perfect accuracy across all graph sizes, with open-source models such as deepseek-R1 and Qwen3-30B also achieving over 97% on small and medium graphs. This suggests that counting connected components is a well-internalized task for many models, especially when the concept is explicitly framed. Notably, the use of "0-dimensional homology" in the prompt does not significantly hinder performance, indicating that many models can align this term with the familiar notion of connectedness.

In contrast, the *1D Simplex Counting* task is considerably more challenging. Even the strongest models fail to surpass 30% accuracy on medium and large graphs. Among them, GPT-4o performs relatively better, but its results remain unsatisfactory. This difficulty likely stems from the inherently discrete nature of counting, as well as the weak association between topological terminology and natural language. Although the term "1-simplex" is mathematically precise, it is rarely used in everyday contexts, and most models may lack the prior knowledge needed to link it reliably to the concept of edges in graphs.

The *Component Reduction* task sits between the two in difficulty. Most models achieve over 95% on small graphs, with Gemini maintaining strong performance even as the graph size increases. The task requires not only understanding the current number of connected components, but also modifying the graph to reduce it by one. This implies a degree of generative reasoning over topological state changes, and the performance gap between models such as Claude (81.5% on large graphs) and Qwen (25.5%) highlights their varying abilities to execute localized structural edits in a topological

*Table 1.* Overview of the LLM4PH Benchmark Tasks.

| Task Category | Task Name | Evaluation | #Samples |
|---|---|---|---|
| Simplicial Structure Understanding | 0D Component Counting | Accuracy | 1200 |
| | 1D Simplex Counting | Accuracy | 1200 |
| | Component Reduction | Accuracy | 1200 |
| Filtration Evolution Reasoning | Component Count Under Filtration | Accuracy | 1200 |
| | Simplex Birth Time | Accuracy | 1200 |
| | Component Merge Time | Accuracy | 1200 |
| Filtration Strategy Design | Optimal Filtration Selection | Ranking | 1200 |
| | Non-Uniform Filtration Generation | Ranking | 1200 |
| Real-World Graph Inference | Filtration Selection for Classification | Accuracy | 900 |
| | Filtration Sequence Generation for Classification | Accuracy | 900 |
| **Total** | 10 | | 11400 |

*Table 2.* Results of Simple Tasks.

| Size | GPT-4.1-mini | GPT-4o | Claude | Gemini | DS-R1 | DS-V3 | Qwen | Mistral | Llama | Random |
|---|---|---|---|---|---|---|---|---|---|---|
| **0D Component Counting** | | | | | | | | | | |
| S | 0.693 | 0.863 | 0.995 | **1** | 0.995 | 0.998 | 0.995 | 0.843 | 0.390 | 0.301 |
| M | 0.518 | 0.733 | 0.915 | **1** | 0.915 | 0.995 | 0.978 | 0.680 | 0.335 | 0.303 |
| L | 0.375 | 0.375 | 0.528 | 0.988 | 0.528 | **0.998** | 0.535 | 0.455 | 0.335 | 0.353 |
| **1D Simplex Counting** | | | | | | | | | | |
| S | 0.213 | 0.270 | 0.150 | **0.271** | 0.150 | 0.260 | 0.143 | 0.085 | 0.223 | 0.253 |
| M | 0.105 | 0.225 | 0.068 | 0.078 | 0.068 | 0.043 | 0.035 | 0.030 | 0.058 | **0.255** |
| L | 0.105 | 0.178 | 0.018 | 0.044 | 0.018 | 0.038 | 0.020 | 0.045 | 0 | **0.265** |
| **Component Reduction** | | | | | | | | | | |
| S | **1** | 0.943 | **1** | **1** | **1** | **1** | 0.998 | 0.898 | 0.875 | - |
| M | 0.993 | 0.855 | 0.988 | **1** | 0.988 | **1** | 0.400 | 0.780 | 0.810 | - |
| L | 0.833 | 0.620 | 0.815 | **1** | 0.815 | **1** | 0.255 | 0.648 | 0.708 | - |

*Table 3.* Results of Medium Tasks.

| Size | GPT-4.1-mini | GPT-4o | Claude | Gemini | DS-R1 | DS-V3 | Qwen | Mistral | Llama | Random |
|---|---|---|---|---|---|---|---|---|---|---|
| **Simplex Birth Time** | | | | | | | | | | |
| S | 0.683 | 0.278 | 0.370 | 0.470 | **0.960** | 0.325 | 0.375 | 0.280 | 0.325 | 0.118 |
| M | 0.375 | 0.135 | 0.415 | 0.125 | **0.810** | 0.140 | 0.135 | 0.130 | 0.345 | 0.143 |
| L | 0.263 | 0.188 | 0.545 | 0.538 | **0.760** | 0.030 | 0 | 0.055 | 0.255 | 0.185 |
| **Component Merge Time** | | | | | | | | | | |
| S | 0.333 | 0.425 | 0.270 | 0.180 | 0.455 | **0.470** | 0.135 | 0.310 | 0.400 | 0.105 |
| M | 0.218 | **0.308** | 0.080 | 0.080 | 0.133 | 0.160 | 0.050 | 0.240 | 0.085 | 0.148 |
| L | **0.260** | 0.258 | 0.013 | 0.027 | 0.050 | 0.150 | 0.030 | 0.150 | 0.010 | 0.133 |
| **Component Count Under Filtration** | | | | | | | | | | |
| S | 0.968 | 0.790 | **0.990** | 0.885 | 0.988 | 0.838 | 0.535 | 0.515 | 0.448 | 0.173 |
| M | 0.958 | 0.593 | 0.885 | 0.810 | **0.995** | 0.660 | 0.448 | 0.422 | 0.240 | 0.120 |
| L | 0.615 | 0.210 | 0.510 | 0.635 | **0.975** | 0.335 | 0.285 | 0.130 | 0.113 | 0.252 |

context.

**Observation.** While many LLMs perform well on classical graph tasks when expressed in plain language, only the strongest models sustain high accuracy when the same tasks are reframed using the abstract vocabulary of PH. **This reveals that true topological reasoning, understood as alignment with the PH pipeline rather than simple output matching, remains an unsolved challenge for most models.**

*Table 4.* Results of Hard Tasks. Subscripts indicate standard deviations.

| Size | GPT-4.1-mini | GPT-4o | Claude | Gemini | DS-R1 | DS-V3 | Qwen | Mistral | Llama |
|---|---|---|---|---|---|---|---|---|---|
| **Optimal Filtration Selection** | | | | | | | | | |
| S | $4.28_{1.75}$ | $4.38_{1.73}$ | $4.05_{1.79}$ | $4.06_{1.93}$ | $4.45_{1.77}$ | $4.22_{1.82}$ | $4.55_{1.73}$ | $3.67_{1.79}$ | $\mathbf{3.58}_{1.74}$ |
| M | $4.14_{1.50}$ | $4.18_{1.65}$ | $4.05_{1.61}$ | $4.38_{1.64}$ | $4.48_{1.52}$ | $4.30_{1.44}$ | $4.30_{1.66}$ | $3.78_{1.55}$ | $\mathbf{3.74}_{1.46}$ |
| L | $4.23_{1.40}$ | $4.24_{1.43}$ | $4.21_{1.43}$ | $4.40_{1.50}$ | $4.46_{1.48}$ | $4.20_{1.46}$ | $4.27_{1.60}$ | $\mathbf{4.05}_{1.49}$ | $4.10_{1.39}$ |
| **Non-Uniform Filtration Generation** | | | | | | | | | |
| S | $46.4_{30.1}$ | $56.0_{38.6}$ | $\mathbf{43.4}_{29.3}$ | $58.3_{34.2}$ | $60.2_{35.4}$ | $54.9_{34.6}$ | $53.7_{34.9}$ | $46.6_{31.5}$ | $50.8_{31.8}$ |
| M | $52.6_{35.1}$ | $54.2_{36.6}$ | $\mathbf{47.9}_{31.5}$ | $54.8_{35.8}$ | $64.4_{35.0}$ | $55.5_{33.5}$ | $59.1_{31.3}$ | $50.0_{33.3}$ | $53.9_{30.4}$ |
| L | $48.6_{33.9}$ | $53.0_{31.0}$ | $46.8_{30.3}$ | $54.7_{33.8}$ | $77.6_{33.5}$ | $54.0_{27.4}$ | $60.9_{34.2}$ | $\mathbf{46.7}_{31.1}$ | $52.6_{28.6}$ |

## 3.2. Medium Tasks: Filtration Evolution Reasoning

**Performance analysis.** Unlike traditional graph reasoning tasks that rely solely on fixed structures, the medium tasks in our benchmark center around filtration. This dynamic process is foundational to PH, as it defines when features appear (birth) and disappear (death). In these tasks, models must go beyond static recognition and reason about how topological features evolve across a filtration sequence. This reflects one of the most distinctive aspects of PH and poses a significantly greater challenge for LLMs, especially without explicit computation.

As shown in Table 3, model performance varies widely across tasks and sizes, highlighting both the complexity of filtration-based reasoning and the limitations of current LLMs.

In the *Simplex Birth Time* task, which requires the model to determine when specific simplices (such as edges or triangles) appear during the filtration process, the results are highly inconsistent. DeepSeek-R1 achieves strong performance on this task (up to 96%), possibly due to its robustness in handling ordering-based reasoning. However, this advantage does not consistently extend to other tasks. In contrast, other open-source models perform poorly. For instance, Qwen reaches only 10% and Mistral just 5.5% on large graphs. These results suggest that reasoning about dynamic structural evolution remains a significant challenge for large language models.

The *Component Merge Time* task, which requires models to determine when two connected components merge during a filtration, is highly challenging for all models. Even top-performing models like GPT-4.1-mini and Claude score below 35%, and some models, such as Claude and Gemini on large graphs, barely produce any correct predictions. Although DeepSeek-R1 achieves the best results on small graphs, its performance declines significantly as the graph size increases. This difficulty further illustrates that tracking the merging of components across filtration steps is a fundamentally hard problem for autoregressive token-based large language models.

On *Component Count Under Filtration*, DS-R1 leads across all sizes with 98.8% on small, 99.5% on medium, and 97.5% on large, showing the strongest scale robustness. Claude is marginally higher on small at 99.0%, and GPT-4.1-mini stays strong on small and medium at 96.8% and 95.8% but drops to 61.5% on large. GPT-4o declines with size from 79.0% to 59.3% to 21.0%. DS-V3 sits mid tier. Open-source models remain far behind, near the mid 50s on small, mid 40s on medium, and below 30% on large, with Llama near 11.0%. Overall, DS-R1 sustains near-ceiling accuracy while others degrade with scale.

**Observation.** Filtration-based reasoning adds a distinct layer of complexity that exposes the structural limitations of current LLMs. While some models manage localized success on small graphs/simple filtrations, none consistently handle the full range of tasks and sizes. **PH introduces not only unfamiliar terminology but also a conceptual framework grounded in temporal evolution and geometric abstraction which remains difficult for most models to grasp.** These results pose a critical challenge in topological reasoning where continuity and change must be understood in tandem.

## 3.3. Hard Tasks: Filtration Strategy Design

**Performance analysis.** This set of tasks evaluates LLMs' ability to reason in reverse: from desired objectives (e.g., maximizing topological difference) to designing appropriate filtration strategies. In contrast to earlier tasks that test a model's ability to recognize topological features, these tasks assess whether a model can actively control the filtration process to induce specific topological outcomes.

We use ranking-based evaluation for both tasks. In *Optimal Filtration Selection*, each model is asked to choose a filtration function (e.g., node degree, edge weight) that maximizes the Wasserstein distance between persistence barcodes of two input graphs. Since there are six candidate functions, the model's answer is ranked among the six: lower rank means better performance (1 is best, 6 is worst). In *Non-Uniform Filtration Generation*, the model generates

a filtration sequence (a length-5 subset from values 1 to 10), and its output is ranked among 126 possible sequences based on the resulting Wasserstein distance. Again, lower rank indicates a more effective choice.

From Table 4, we observe that all models struggle with these tasks. In the *Optimal Filtration Selection* task, the small open-source models Mistral and Llama surprisingly achieve the best results, while most other models have average ranks above 3. In addition, all models exhibit large standard deviations, indicating highly unstable outputs. This suggests that their performance is only marginally better than random guessing. Overall, designing filtrations based on specific goals and graph structures remains a highly challenging task for current language models.

In the *Non-Uniform Filtration Generation* task, Claude emerges as the best-performing model across small and medium graph sizes, achieving the lowest average rank. This indicates a surprising strength in generating effective filtration sequences that yield topological divergence. However, the overall performance remains disappointing. All models exhibit large standard deviations that are nearly as large as their means, suggesting that the models have not truly learned any meaningful knowledge related to the task.

**Observation.** Topological control tasks remain challenging for LLMs, requiring reasoning about graph-filtration interactions. Our benchmark tests active topological shaping rather than passive interpretation. Results show that even strong models lack sufficient reasoning granularity, suggesting new research directions in persistence-aware training and hybrid architectures. **Success requires aligning structural patterns with topological objectives, not just pattern recognition.**

### 3.4. Results of Real-World Tasks (or Real-World Graph Inference)

This set of tasks evaluates whether LLMs can apply PH to graph classification. It includes two PH-guided tasks: *Filtration Selection for Classification* and *Filtration Sequence Generation for Classification*, as well as a direct classification baseline without topological reasoning.

In the *Filtration Selection for Classification* task, performance is notably poor and highly uniform across models and datasets, with accuracies fluctuating narrowly around 0.32 to 0.36. This indicates that models consistently choose similar filtration functions regardless of the graph structure. Further inspection reveals that most LLMs default to selecting `edge weight` as the filtration function, possibly because it appears more "interpretable" or statistically grounded. However, this uniformity fails to expose discriminative topological differences, leading to almost random classification results.

In contrast, the *Filtration Sequence Generation for Classification* task exhibits stronger performance across all models, especially on the LDHFR dataset, where top models (Claude, GPT-4.1-mini, Gemini) achieve accuracies above 0.95. This suggests that when explicitly instructed to manipulate filtration sequences, even under fixed filtration functions, LLMs can induce more topologically meaningful separations. Notably, Claude outperforms all other models on BZR and LDHFR, achieving the best classification accuracy among all methods.

To evaluate the reliance on PH reasoning, we introduce a control task: *Direct Classification*, where the model is asked to classify graphs directly without designing filtration strategies. Surprisingly, several strong models achieve near-perfect accuracy (0.98 to 1.0), suggesting that for small 4-graph inputs, LLMs can leverage shallow statistical patterns or implicit textual correlations to memorize or infer class labels. However, this also reveals that the PH tasks are substantially more difficult than the direct version, requiring abstraction, planning, and topological insight rather than pattern matching.

**Observation.** The weak performance on filtration function selection, contrasted with stronger results on filtration sequence generation, suggests that LLMs are capable of expressing meaningful topological reasoning **only when guided by structured prompts and constrained options**. In the absence of such scaffolding, models tend to fall back on familiar heuristics, which often obscure rather than uncover PH-based structure.

## 4. Cross-Task Analyses

Beyond the evaluation of individual tasks, we conduct several cross-task experiments that focus on broader aspects of model behavior. These experiments examine the effects of input representation, model fine-tuning, and pipeline compositionality across multiple tasks and graph sizes. Together, they provide additional insight into the generalization and operational behavior of language models when applied to PH reasoning. Detailed results for all cross-task experiments are provided in Appendix F.

**Prompt Encoding and Task Formulation Ablation**

We examine how prompt structure affects model behavior by varying both the **graph representation style** (text-style, code-like, matrix-based) and the **task formulation style** (topological, graph-theoretic, minimal) across three representative tasks on synthetic and real-world graphs. As shown in Table 11, text-style inputs consistently lead to higher accuracy, and graph-theoretic phrasing generally outperforms topological or minimal formulations. In contrast, matrix-based and minimal prompts yield lower and less stable performance, confirming that surface-level prompt

*Table 5.* Results of Real-World Tasks.

| Datasets | GPT-4.1-mini | GPT-4o | Claude | Gemini | DS-R1 | DS-V3 | Qwen | Mistral | Llama |
|---|---|---|---|---|---|---|---|---|---|
| **Filtration Selection for Classification** | | | | | | | | | |
| BZR | **0.350** | **0.350** | **0.350** | 0.340 | 0.320 | **0.350** | 0.320 | **0.350** | **0.350** |
| COX2 | 0.320 | 0.260 | 0.320 | 0.320 | **0.360** | 0.320 | **0.360** | 0.290 | 0.315 |
| LDHFR | 0.345 | 0.310 | 0.345 | 0.345 | **0.460** | 0.335 | **0.460** | 0.310 | 0.340 |
| **Filtration Sequence Generation for Classification** | | | | | | | | | |
| BZR | 0.595 | 0.650 | **0.775** | 0.765 | 0.760 | 0.410 | 0.400 | 0.652 | 0.740 |
| COX2 | 0.710 | 0.555 | 0.695 | **0.755** | 0.180 | 0.160 | 0.480 | 0.655 | 0.655 |
| LDHFR | 0.950 | 0.920 | **0.970** | 0.958 | 0.760 | 0.880 | 0.740 | 0.860 | 0.945 |
| **Direct Classification** | | | | | | | | | |
| BZR | 0.985 | 0.940 | 0.990 | 0.969 | 0.920 | 0.980 | 0.820 | 0.525 | **0.995** |
| COX2 | 0.985 | 0.920 | 0.995 | 0.990 | 1 | 0.995 | 0.740 | 0.460 | 1 |
| LDHFR | 1 | 0.960 | 0.995 | 1 | 1 | 0.990 | 0.980 | 0.790 | 1 |

features substantially influence outcomes even when the underlying task remains unchanged.

To further explore formulation effects, we conduct a **terminology ablation study** comparing paired prompts using topological terms (e.g., H0 class) versus graph-theoretic terms (e.g., connected component) with identical task logic. Results (Table 12) show task-dependent sensitivity, for instance, GPT-4o performs similarly across terms on the birth task but drops significantly on the CCUF task when using PH-specific terminology. This indicates that while terminology contributes to variance, many failures arise from deeper reasoning limitations.

Finally, we test whether prompt engineering can mitigate these failures in the non-uniform filtration generation task using Few-shot Chain-of-Thought (CoT) and rule-emphasized prompting. Neither approach yields consistent gains across models (Table 13), suggesting that prompt design alone is insufficient to resolve challenges in multi-step topological reasoning, particularly transitions between homology classes.

**Post Training and Scale Transferability** We study whether supervised adaptation can instill stable topological routines in an open-source model. We fine tune LLaMA-3.1-8B-Instruct on small graphs for three representative tasks and then evaluate on medium graphs with matched distributions. The protocol preserves the prompt settings used in the ablation, which allows us to measure how much performance remains sensitive to surface representation after adaptation. As reported in Table 14, post training delivers consistent and often large gains across all tasks. On 1D Simplex Counting the best prompt improves from 0.02 to 0.20, while Code like with topological phrasing reaches 0.26. On Component Merge Time the score rises from 0.09 to 0.43 for text with graph theoretic phrasing, with similar improvements for other text prompts. On Component Count Under Filtration

the text with graph theoretic phrasing increases from 0.16 to 0.66, and code like with topological phrasing reaches 0.54. Prompt sensitivity remains present after adaptation. Text prompts continue to be reliable across tasks. Code like prompts become more competitive after fine tuning and show the largest relative improvements in two tasks. Minimal phrasing improves but still trails behind explicit instructions. We observe the same trends in cross scale transfer, where a model tuned on small graphs improves from 0.005 to 0.235 on large 1D Simplex Counting, which suggests that the learned routines generalize beyond the training size. Overall, fine tuning strengthens topological reasoning and reduces variance, yet careful prompt design remains important for peak performance.

**Compositional PH Pipeline** Beyond evaluating LLMs on individual subtasks, we construct a full pipeline to assess their ability to guide an integrated persistent homology (PH) workflow. Using the real-world BZR dataset, we run a *Filtration Selection for Classification* experiment following a standard five-step PH process: (1) select a filtration function, (2) compute filtration values, (3) generate persistence diagrams, (4) compare diagram distances, and (5) determine the final class label. As a reference, we also report results from standard PH tools using four common filtration functions: Weight, Degree, Betweenness, and Closeness.

Table 9 compares four hybrid settings that vary the boundary between model reasoning (`LLM`) and tool-based execution (`code`). For example, in Setting 1, the model handles filtration selection, diagram generation, and distance comparison, while the remaining steps are delegated to code; other settings shift the diagram or distance steps back to code. We find that the pure code-based pipeline, with only filtration selection done by the model, yields stable mid-level accuracy across models. Interestingly, hybrid configurations where the model performs distance comparison produce the highest scores for Claude and DS-V3. In contrast, settings

that require models to generate diagrams lead to performance degradation, underscoring the difficulty of symbolic diagram construction via language prompts.

Overall, these results suggest that LLMs are most effective at filtration selection and, in some cases, at interpreting diagram-level similarities, while code remains critical for stable and accurate diagram generation. While PH tools consistently yield the highest accuracy, certain hybrid configurations allow models like Claude to match or even exceed tool-only performance, reaching up to 0.840 accuracy. This illustrates the promise of hybrid neuro-symbolic pipelines and highlights filtration selection as a particularly viable interface for language model integration.

**Large-Graph Scalability via Subgraph Sampling** Since current context windows cannot contain full large graphs, we evaluate scalability through controlled subgraph sampling on the ego-Facebook dataset (Leskovec & Mcauley, 2012). We use random walk sampling to build six collections with 10, 20, 50, 80, 100, and 150 subgraphs, each subgraph containing 50 nodes. We run the 1D Simplex Counting task five times for GPT-4o and GPT-4.1-mini and report mean accuracy (standard deviation) in Table 10. Both models improve as the number of sampled subgraphs grows. GPT-4.1-mini increases monotonically from 14% to 24%, and GPT-4o shows an overall upward trend.

**Error analysis: Non-Uniform Filtration Generation** To better understand model failure on this task, we conduct a systematic error analysis and identify six recurring patterns for the Non-Uniform Filtration Generation task. These fall into three broad categories: loss of prior context, incomplete procedural reasoning, and chain-of-thought instability.

1. *Fact forgetting.* The model fails to retain previously mentioned edge information. For instance, it may first note that node 1 is connected to node 7, but later treat them as belonging to separate components. This occurs because earlier facts drift out of the model's attention window and are no longer integrated into downstream predictions.

2. *Rule forgetting.* The model may explicitly acknowledge rules, such as "isolated nodes are not counted as components" or "a merge occurs when two components join," but fail to enforce them later. This reflects the absence of persistent logical constraints; once the model's focus shifts, prior rules are no longer enforced.

3. *Incomplete algorithm emulation.* The model often attempts to reproduce the output of traversal-based algorithms (such as depth-first search) but does so heuristically, without internal state tracking. This results in invalid outputs, such as calling a walk with repeated nodes a cycle or skipping key edges during merge detection.

4. *Blurry conceptual boundaries.* Topological concepts are applied too loosely. For example, any path resembling a loop may be labeled as a "hole" or "cycle," even if it includes repeated vertices or fails basic structural criteria. This suggests the model relies on surface-level statistical cues rather than precise graph-theoretic definitions.

5. *Interrupted reasoning.* In longer examples requiring multiple stages of analysis, the model's output may terminate mid-thought (e.g., "Now we check for cycles..." with no continuation). This typically occurs when the chain of thought becomes too long to maintain internal coherence, especially in graphs with many nodes or edge weights.

6. *Analysis paralysis.* When faced with multiple plausible merge paths or interpretations, the model may enter loops of self-revision. For instance, it might state "the correct answer is 4.00," then immediately consider "perhaps 10.00," and continue revising. This failure to converge indicates difficulty in managing uncertainty and resolving conflicting hypotheses.

These failure modes reveal structural limitations in current LMs. Robust reasoning under non-uniform filtrations needs persistent memory, explicit constraint handling, and the ability to simulate structured procedures.

## 5. Conclusion

We present LLM4PH, a benchmark for evaluating whether large language models can reason about global structural evolution in graphs through persistent homology. By decomposing the PH pipeline into interpretable tasks from structural recognition to filtration strategy design, the benchmark separates local graph understanding from reasoning over filtration driven state changes. The results show that current models are not uniformly weak on PH tasks. They can succeed on local or strongly scaffolded settings, but they often fail when they must choose an abstract filtration strategy or maintain global connectivity state across multiple thresholds. The contrast between weak filtration selection and stronger fixed strategy sequence generation suggests a specific limitation in declarative topological planning, while the frequent preference for edge weight filtrations points to a surface heuristic that can obscure more discriminative graph structure. These findings position LLM4PH as a testbed for studying structure-aware scientific reasoning.

## Acknowledgments

This work was supported by the Key R&D Program of Hubei Province (Project No. JSCX202500180) and the Key R&D Program of Wuhan City (Project No. 2025050602030055). Chen and Gel have received no research support from any non-US-based entity.

## Impact Statement

This work contributes a principled benchmark for evaluating LLMs' ability to reason about structural evolution in graphs. By focusing on persistent homology as a testbed for multi-scale reasoning, it advances our understanding of LLMs' limitations in topological abstraction. The benchmark may inform future research in scientific AI, structure-aware reasoning, and hybrid symbolic–neural systems. No foreseeable societal risks are associated with this work.

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

## A. Related Work

**LLM for Graphs**   Recent efforts have sought to evaluate the reasoning capabilities of large language models (LLMs) on graph-structured data through benchmark datasets. Early benchmarks focus on static graph tasks such as shortest path computation, cycle detection, node classification, and graph traversal (Fatemi et al., 2024; Wang et al., 2023). More recent work expands this scope to dynamic graphs, introducing temporal reasoning tasks such as dynamic link prediction and event ordering (Zhang et al., 2024). Other studies have explored graph pattern recognition (Dai et al., 2025) and subgraph isomorphism through natural language prompts. These benchmarks typically emphasize local structure, edge-level logic, or symbolic reasoning over discrete structures. However, they often overlook the global, multi-scale, and high-dimensional characteristics inherent in many real-world graphs. Our work complements and extends this line of research by introducing PH as a testbed for evaluating topological reasoning beyond pairwise relationships.

Recent work has explored using LLMs for traditional graph learning tasks (Chen et al., 2024c). Methods like Graph-LLM (Chai et al., 2025) and GraphText (Zhao et al., 2024) convert graphs into natural language for few-shot inference, while others combine LLMs with GNNs (Tang et al., 2024; Tan et al., 2025). Graph instruction tuning methods such as GraphWiz (Chen et al., 2024a) and InstructGraph (Wang et al., 2024) further improve LLM behavior on graph computational tasks. However, these approaches focus on low-order features rather than high-dimensional topological structures. LLM4PH deliberately evaluates text based graph interfaces so that graph textualization remains part of the reasoning problem. Graph encoder coupled LLMs address a complementary question concerning whether a stronger graph interface can reduce the observed topological reasoning gap.

**Persistent Homology on Graphs**   Persistent homology, a core tool in topological data analysis (TDA), provides a framework to extract multi-scale topological features from data (Edelsbrunner & Harer, 2010; Zomorodian & Carlsson, 2005; Carlsson, 2009; Carlsson & Gabrielsson, 2020; Pun et al., 2022). When applied to graphs, it captures global structures such as connected components, cycles, and voids across a filtration of the graph (Horak et al., 2009; Su et al., 2025). Persistent homology on graphs has been widely used in fields like neuroscience (Dabaghian et al., 2012; Curto & Sanderson, 2025), biology (Xia & Wei, 2014; Meng et al., 2020; Townsend et al., 2020), materials science (Obayashi et al., 2022), and blockchain analytics (Akcora et al., 2020; Ofori-Boateng et al., 2021; Chen et al., 2022; 2025) often serving as a shape descriptor or structural feature for downstream tasks. In machine learning, topological features have been integrated as a fully trainable layer into graph neural networks (Hofer et al., 2019; Carlsson & Gabrielsson, 2020) as well as most recently used within graph contrastive learning and metalearning frameworks (Chen et al., 2024b; Zheng et al., 2026; Li et al., 2025) and graph diffusion (Chen & Gel, 2025; Park et al., 2026; Huang & Birdal, 2026). Despite its growing adoption, prior studies largely assume full access to the topological pipeline and does not explore whether LLMs can internalize or reason over such structures directly. Our work bridges this gap by proposing a benchmark that tests whether LLMs can understand, predict, and generate persistent homology concepts from graph inputs without explicit computation.

**Simplicial Complex Identification**   Recent efforts in understanding and identifying higher-order topological structures have extended classical graph methods to the realm of hypergraphs and simplicial complexes. Babai and Codenotti (Babai & Codenotti, 2008) analyzed the complexity of hypergraph isomorphism, showing that while the general problem is NP-complete, isomorphism between bounded-rank hypergraphs, such as low-dimensional simplicial complexes, admits moderately exponential-time algorithms via group-theoretic methods. To capture structural similarity beyond exact isomorphism, recent work has introduced polynomial-time heuristics based on combinatorial refinement. Feng et al. (Feng et al., 2024) proposed a generalization of the Weisfeiler–Lehman (WL) test to hypergraphs and constructed WL-based hypergraph kernels, enabling efficient comparisons between high-order structures through subtree and hyperedge patterns. Building on this, Zhang et al. (Zhang et al., 2025) reinterpreted such kernels in terms of homomorphism counts, revealing that many existing methods implicitly focus on acyclic patterns and introducing the Subtree-Cycle Kernel to enhance expressiveness by capturing cyclic features. These approaches provide both theoretical insights and practical tools for identifying and comparing simplicial complexes through a combination of isomorphism heuristics and homomorphism-informed embeddings.

## B. Dataset Construction

To support a controlled and interpretable evaluation of persistent homology understanding, we construct both synthetic and real-world datasets tailored to each task category.

## B.1. Benchmark Statistics

The statistical overview of our benchmark is illustrated in Table 1. Each synthetic dataset task category's samples are composed of an equal number of small, medium, and large graphs (10 nodes, 15 nodes, 30 nodes). For example, out of 1200 samples, 400 are small graphs, 400 are medium graphs, and 400 are large graphs. The real-world datasets BZR, COX2, and LDHFR have an equal number of samples.

## B.2. Synthetic Graph Generation

All synthetic graphs are generated using randomized network models with fixed size constraints. Specifically, small, medium, and large graphs are set to contain 10, 15, and 30 nodes respectively. For each graph, we randomly sample edges to ensure structural variability while maintaining connectivity and topological relevance.

## B.3. Task-specific Filtering and Selection

For **simple** and **medium** tasks, we compute the Vietoris–Rips complex from each candidate graph and evaluate its corresponding Betti numbers and persistence diagrams across standard filtration functions. Graphs are selected based on whether they yield diverse and interpretable topological outcomes, such as distinct connected components, nontrivial 1-dimensional features, and clear merging events.

For **hard tasks**, we compute pairwise Wasserstein distances between persistence diagrams of graph pairs under all candidate filtration strategies. Each strategy (e.g., node degree, edge weight, closeness centrality) is evaluated, and we retain the ranking of each strategy based on the resulting distance. This ranking provides the ground truth supervision for tasks involving filtration design.

## B.4. Real-world Graph Selection

For **real-world tasks**, we curate graphs from public datasets (e.g., BZR, COX2, LDHFR) and sample small subsets of graphs from two distinct classes. For each problem instance, we evaluate the Wasserstein distance between diagrams produced under each candidate filtration strategy, and label tasks according to whether the correct class separation can be achieved. Graphs are allowed to appear in at most two tasks to avoid overfitting and ensure evaluation diversity.

This construction pipeline ensures that each benchmark task is grounded in verifiable topological variation and aligned with persistent homology computation standards.

# C. Additional Explanation of Persistent Homology

Persistent homology is a core method in topological data analysis (TDA) that enables the extraction of multi-scale topological features from data. It generalizes classical homology theory by introducing the concept of a *filtration*, which allows one to track how topological features evolve across scales.

A filtration is a nested sequence of simplicial complexes:

$$K_0 \subseteq K_1 \subseteq \cdots \subseteq K_T,$$

typically constructed by gradually adding simplices based on a filtration function (e.g., edge weights in a graph). As the filtration evolves, features such as connected components (0-dimensional homology), cycles (1-dimensional), and voids (2-dimensional) appear and eventually disappear.

For example, in a social network where edge weights reflect communication frequency, a filtration can be constructed by adding edges from the weakest to the strongest ties. As shown in Figure 1, graphs $G_a$ and $G_b$ differ only in the weight of a single edge, yet their topological structures evolve differently. Specifically, graph $G_a$ maintains a single connected component throughout the filtration, while graph $G_b$ initially has two separate components that eventually merge into one. PH captures these structural differences concisely, producing barcodes that summarize when components appear, persist, and merge across different interaction thresholds. This helps distinguish stable community structures from transient interactions and noise.

Key terms used throughout this work include:

- **Simplex**: A generalization of a vertex (0-simplex), edge (1-simplex), triangle (2-simplex), or higher-dimensional face. Simplices are the building blocks of a simplicial complex.

- **Simplicial Complex**: A finite set of simplices closed under the subset operation. It defines a discrete topological space amenable to homology computation.

- **Filtration Function**: A function $f$ that assigns a real-valued threshold to each simplex (often indirectly through edges or vertices). The function governs the order in which simplices are added during the filtration.

- **Birth and Death**: A topological feature (e.g., a connected component or a cycle) *births* at the filtration index where it first appears and *dies* when it is merged into a larger feature or filled in.

- **Persistence Diagram / Barcode**: A multiset of intervals $\{[b_i, d_i]\}$ representing the lifespan of topological features. The length $d_i - b_i$ reflects the *persistence* or significance of each feature.

- **Betti Number** $\beta_k$: The rank of the $k$-th homology group $H_k$, counting $k$-dimensional holes: $\beta_0$ for components, $\beta_1$ for loops, etc.

- **Wasserstein Distance**: A distance metric between persistence diagrams that quantifies how topological structures differ across filtrations. It is often used to evaluate the output of persistence-aware models.

By computing and analyzing these topological signatures, persistent homology provides a robust, noise-tolerant summary of data structure that is invariant to continuous transformations and particularly suited to graph-structured domains.

## D. Details of Task Design

This appendix provides detailed explanations for each task in the LLM4PH benchmark, framed through the mathematical language of persistent homology and simplicial complex theory.

### D.1. Simple Tasks (Simplicial Structure Understanding)

These tasks correspond to the initial stage of persistent homology, prior to any filtration, where the simplicial structure of the graph must be identified. Each task tests whether an LLM can understand basic homological objects from a combinatorial graph input.

**0D Component Counting.** Given an undirected graph $G = (V, E)$, the model is asked to compute $\beta_0(G)$, the 0-th Betti number, which equals the number of connected components in the graph. This is equivalent to computing the rank of the 0-dimensional homology group $H_0(G)$.

**1D Simplex Counting.** Given a graph $G$, the task requires counting the number of 1-simplices, which correspond to edges in the 1-skeleton of the simplicial complex induced by $G$. This measures the cardinality of the 1-simplex set $\Sigma_1$.

**Component Reduction.** Given $G$, the model is asked to identify a single edge $e \notin E$ such that $E' = E \cup \{e\}$ decreases $\beta_0(G)$ by one. This requires reasoning about connected component merges via edge addition.

### D.2. Medium Tasks (Filtration Evolution Reasoning)

These tasks test an LLM's ability to reason over a filtration $\{K_t\}_{t \in T}$ of simplicial complexes built from a weighted graph $G$, where $K_t$ denotes the simplicial complex at filtration threshold $t$. The aim is to understand topological evolution: the birth and death of homology classes across $t$.

**Simplex Birth Time.** Given a weighted graph $G$ and a filtration order (e.g., by edge weight), the model predicts the value $t$ at which a given simplex $\sigma \in K_t$ appears. This corresponds to the birth time $b(\sigma)$ in the persistence diagram.

**Component Merge Time.** For a pair of initially disconnected vertices $(u, v)$, the model predicts the threshold $t$ at which $u$ and $v$ are first included in the same connected component, i.e., when their representatives in $H_0$ are merged.

**Component Count Under Filtration.**    At a given threshold $t$, the model is asked to compute $\beta_0(K_t)$, the number of connected components in the complex $K_t$.

### D.3. Hard Tasks (Filtration Strategy Design)

These tasks involve reasoning from desired topological divergence toward filtration design. Given two graphs $G_1$ and $G_2$, the objective is to select or generate a filtration function $f : E \rightarrow \mathbb{R}$ (or a sequence of filtration thresholds) that induces persistence diagrams $D_1, D_2$ with maximal Wasserstein distance $W(D_1, D_2)$.

*Table 6.* Candidate filtration functions used in filtration selection tasks.

| Filtration function | Structural interpretation |
|---|---|
| Edge weight | Adds edges according to observed pairwise strength or distance and captures how connectivity changes across weighted relations. |
| Node degree | Prioritizes locally connected nodes and reveals how hub centered connectivity shapes component growth. |
| K-shell | Orders nodes by their position in the core periphery structure and emphasizes the emergence of dense graph cores. |
| Closeness centrality | Prioritizes nodes with short average paths to the rest of the graph and captures globally central regions. |
| Betweenness centrality | Prioritizes nodes that lie on many shortest paths and highlights bridges between communities. |
| Eigenvector centrality | Prioritizes nodes connected to other influential nodes and emphasizes recursively reinforced connectivity. |

**Optimal Filtration Selection.**    The model selects a global filtration function $f \in \mathcal{F}$ from a predefined set listed in Table 6 such that the resulting persistence diagrams maximize $W_p(D_1, D_2)$ under a $p$-Wasserstein metric.

**Non-Uniform Filtration Generation.**    Under a fixed filtration function $f$, the model selects a non-uniform filtration sequence $\{t_1, t_2, \ldots, t_5\}$ from a discrete set (e.g., $\{1, 2, \ldots, 10\}$) to maximize $W(D_1, D_2)$, effectively constructing a coarse but topologically expressive filtration.

### D.4. Real-World Tasks (Real-World Graph Inference)

These tasks examine whether an LLM can apply persistent homology insights to downstream tasks such as graph classification. Let $\mathcal{G} = \{G_1, G_2, G_3, G_4\}$ be a set of graphs belonging to two classes. The goal is to design a filtration such that the resulting persistence-based representations $\{D_i\}$ allow for correct class separation.

**Filtration Selection for Classification.**    The model selects a filtration function $f \in \mathcal{F}$ that yields persistence diagrams best aligned with the class partition of the graphs, typically optimizing for inter-class distance or clustering.

**Filtration Sequence Generation for Classification.**    Under a given filtration function $f$, the model generates a filtration sequence $\{t_1, \ldots, t_5\}$ that maximizes topological separability between graphs from different classes, typically measured by diagram-level distance or homology-aware clustering accuracy.

# E. Prompt Examples

This section gives representative prompt templates used in LLM4PH. The placeholders `<GRAPH>` and `<GRAPH 1>`, `<GRAPH 2>` are replaced by graph descriptions generated from the corresponding benchmark instance. The default graph representation is an edge list. The prompt ablation replaces this representation with matrix based or code like graph descriptions while keeping the task target fixed. The listing content is shown verbatim to document the actual evaluation inputs, including their original formatting.

*Listing 1.* Example prompt for 0D Component Counting.

```
You are a mathematical expert specializing in graph theory and persistent homology. Given the following graph
    structure:
Graph Structure:
<GRAPH>

Please calculate the number of connected component in this graph(vertex that not connected to other vertex is not
    a connected component).
And strictly answer in following format:
```

```
Answer:
    connected components: n
(e.g.
Answer:
    connected components: 3
```

*Listing 2.* Example prompt for Component Merge Time with graph theoretic wording.

```
You are an analytical reasoning assistant.

Please calculate features based on the following graph structure.
Graph Structure:
<GRAPH>

Task: Determine the first merging time of graph components.
You can follow these steps:
Step1: Activate edges in the graph in ascending order of their weights and track the resulting graph components.
Step2: Identify the edge with the smallest weight that causes two previously separate components to merge.
Step3: The first merging time t corresponds to the weight of this merging edge.

Please answer in the following format:
Answer:
    death time:[t]
(e.g.
Answer:
    death time:[4])
Please ensure final answer strictly follows above format.
```

*Listing 3.* Example prompt for Non-Uniform Filtration Generation.

```
You are a mathematical expert specializing in graph theory and persistent homology. Given the following two graph
    structures:

graph1 structure:
<GRAPH 1>

graph2 structure:
<GRAPH 2>

Task: Please select a filtration value sequence from [1, 2, ..., 10] that maximizes the difference between graph 1
    and graph 2(maximizes the Wasserstein distance between their persistence barcodes).

You can follow these steps:
Step1: Compare the structure of graph1 and graph2 to see which is denser, whether there are cycles, etc.
Step2: From the given filtration values, identify values that trigger major topological changes in the graphs.
Step3: Choose 5 filtration values that maximize the difference in persistence barcodes between the two graphs(the
    max filtration value should be 10).

Please answer in the following format:
Answer:
    filtration value: [filtration value]
(e.g.
Answer:
    filtration value: [1,3,4,7,10]
)
Please ensure your answer strictly follows this format.
```

*Table 7.* Prompt variants used in representation and terminology ablations.

| Variant | Example fragment |
| --- | --- |
| Text style | `Graph with 4 nodes and 3 edges.  Node 0-[1.00]-Node 1.` |
| Matrix based | `Graph with 4 nodes as an adjacency matrix.  Entry (i,j) gives the edge weight.` |
| Code like | `Graph [name="G"]{ node_list=[...]; edge_list=[...]; }` |
| Topological wording | `Compute the first H0 death time in the filtration.` |
| Graph theoretic wording | `Determine the first merging time of graph components.` |
| Minimal wording | `Count connected components.  Return only the answer.` |

# F. Supplementary Tables for Cross-Task Analyses

This appendix provides supplemental materials to support the results presented in the Sec. 4. We include:

- **Prompt Examples (Appendix E)**, providing representative task prompts and prompt ablation variants.

- **Prompt Style Summary (Table 15)**, outlining the different ways graph structure and task instructions are encoded.

- **Prompt Ablation Results (Table 11)**, showing performance across multiple representation and instruction combinations.

- **Post-Training Evaluation (Table 14)**, reporting transfer results after supervised tuning on small graphs.

- **GraphWiz Baseline (Table 8)**, testing whether graph specialized instruction tuning transfers to PH reasoning tasks.

- **Compositional Pipeline Evaluation (Table 9)**, examining model performance in hybrid workflows across persistent homology steps.

- **Subgraph Scaling (Table 10)**, assessing accuracy trends as the number of sampled subgraphs increases.

### F.1. GraphWiz Baseline

We evaluate a graph specialized baseline starting from the GraphWiz LLaMA2 7B DPO checkpoint and adapt it to the LLM4PH task format with LoRA using 700 task format dialogue instances. This comparison is intended to test whether graph instruction tuning transfers to PH reasoning under the same text based interface. The model produces format compliant outputs on 0D Component Counting and Component Merge Time. It shows partial or complete format failure on Non-Uniform Filtration Generation and Direct Classification, so those results are reported for completeness. For ranking metrics, 127.0 denotes invalid outputs assigned one rank worse than the 126 valid filtration sequences.

*Table 8.* GraphWiz baseline results on representative LLM4PH tasks.

| Task | GraphWiz | Best in paper | Output validity |
|---|---|---|---|
| 0D Component Counting | 0.110 | 1.000 | Format compliant |
| Component Merge Time | 0.067 | 0.470 | Format compliant |
| Non-Uniform Filtration Generation | 127.0 | 43.4 | Partial format failure |
| Direct Classification on BZR | 0.000 | 0.995 | Format failure |

On the two tasks with valid output format, GraphWiz remains far below the strongest general LLMs. This gap suggests that graph instruction tuning on standard graph computational problems does not directly provide the filtration based state tracking and strategy design required by LLM4PH.

*Table 9.* Compositional PH pipeline on BZR. The model chooses the filtration in every row.

| Settings | | | | | Accuracy | | | | |
|---|---|---|---|---|---|---|---|---|---|
| Filtration selection | Filtration values | Diagram generation | Distance comparison | Classifier | GPT-4o | GPT-4.1-mini | Claude | DS-V3 | Mistral |
| LLM | CODE | LLM | LLM | CODE | 0.150 | 0.180 | 0.780 | 0.290 | 0.210 |
| LLM | CODE | CODE | CODE | CODE | 0.350 | 0.350 | 0.350 | 0.350 | 0.350 |
| LLM | CODE | CODE | LLM | CODE | 0.250 | 0.280 | 0.840 | 0.780 | 0.520 |
| LLM | CODE | LLM | CODE | CODE | 0.300 | 0.320 | 0.190 | 0.025 | 0.150 |
| WEIGHT | CODE | CODE | CODE | CODE | | | 0.790 | | |
| DEGREE | CODE | CODE | CODE | CODE | | | 0.610 | | |
| BETWEENNESS | CODE | CODE | CODE | CODE | | | 0.890 | | |
| CLOSENESS | CODE | CODE | CODE | CODE | | | 0.800 | | |

*Table 10.* Subgraph sampling on ego–Facebook for 1D Simplex Counting. Values are mean accuracy $\pm$ standard deviation over five runs.

| Subsamples | 10 | 20 | 50 | 80 | 100 | 150 |
|---|---|---|---|---|---|---|
| GPT–4o | 0.0600±0.08 | 0.1400±0.08 | 0.1280±0.04 | 0.1325±0.03 | 0.1733±0.02 | 0.1960±0.02 |
| GPT–4.1–mini | 0.1400±0.10 | 0.1400±0.08 | 0.1520±0.03 | 0.1850±0.03 | 0.2260±0.02 | 0.2413±0.04 |

*Table 11.* Prompt encoding and task formulation ablation shown in a single horizontally merged table across three tasks.

| | 1D Simplex Counting | | | | Component Merge Time | | | | Component Count Under Filtration | | | |
|---|---|---|---|---|---|---|---|---|---|---|---|---|
| | Synthetic | | LDHFR | | Synthetic | | LDHFR | | Synthetic | | LDHFR | |
| Prompt setting | GPT-4o | Claude | GPT-4o | Claude | GPT-4o | Claude | GPT-4o | Claude | GPT-4o | Claude | GPT-4o | Claude |
| Text style + Graph theoretic | **0.270** | 0.150 | 0.170 | 0.260 | 0.425 | 0.270 | **0.250** | 0.090 | 0.785 | **0.983** | 0.173 | **0.780** |
| Text style + Minimal | 0.176 | 0.078 | 0.273 | **0.670** | 0.443 | 0.225 | 0.160 | 0.023 | 0.093 | 0.025 | 0.233 | 0.085 |
| Text style + Topological | 0.160 | 0.268 | 0.250 | 0.250 | **0.510** | 0.090 | 0.125 | 0.020 | 0.633 | 0.790 | 0.148 | 0.655 |
| Matrix style + Topological | 0.125 | 0.224 | 0.118 | 0.230 | 0.245 | 0.090 | 0.018 | 0.000 | 0.643 | 0.870 | 0.180 | 0.610 |
| Code like + Topological | 0.128 | 0.210 | 0.118 | 0.160 | 0.445 | 0.115 | 0.220 | 0.110 | 0.833 | 0.980 | 0.120 | 0.570 |

*Table 12.* Terminology ablation results comparing topological vs. graph theoretic phrasing across three tasks.

| Condition | GPT-4.1-mini | GPT-4o |
|---|---|---|
| Graph terminology birth | 0.67 | 0.33 |
| Topology terminology birth | 0.6533 | 0.3233 |
| Graph terminology merge | 0.02 | 0.5533 |
| Topology terminology merge | 0.0066 | 0.57 |
| Graph terminology CCUF | 0.99 | 0.8633 |
| Topology terminology CCUF | 1.0 | 0.45 |

*Table 13.* Performance on non uniform filtration generation with different prompting strategies (Mean Rank ± std)

| Prompt Type | GPT-4o | Claude |
|---|---|---|
| Few shot CoT | 52.09 ± 32.84 | 48.76 ± 29.99 |
| Rule emphasized | 53.11 ± 34.86 | 40.17 ± 31.31 |

*Table 14.* Post training on LLaMA-3.1-8B-Instruct with supervision on small graphs and evaluation on medium graphs.

| Prompt setting | 1D Simplex Counting | | Component Merge Time | | Component Count Under Filtration | |
|---|---|---|---|---|---|---|
| | Zero-shot | Post-train | Zero-shot | Post-train | Zero-shot | Post-train |
| Text style + Graph theoretic | 0.020 | 0.200 | 0.090 | 0.430 | 0.160 | 0.660 |
| Text style + Minimal | 0.000 | 0.160 | 0.040 | 0.290 | 0.090 | 0.400 |
| Text style + Topological | 0.000 | 0.040 | 0.080 | 0.430 | 0.330 | 0.480 |
| Matrix style + Topological | 0.000 | 0.060 | 0.140 | 0.250 | 0.210 | 0.480 |
| Code like + Topological | 0.000 | 0.260 | 0.010 | 0.370 | 0.220 | 0.540 |

*Table 15.* Summary of graph representation and task instruction styles used in the prompt ablation.

| Prompt Style | Description |
| --- | --- |
| *Graph Representation Styles* | |
| Text-style format | Nodes and edges are described using plain natural language sentences, following human-like narration. |
| Code-like format | Graph elements are structured using syntax that resembles programming or configuration files, such as lists or dictionaries. |
| Matrix-based format | The graph is represented by a numerical adjacency matrix, requiring structural inference from tabular data. |
| *Task Instruction Styles* | |
| Topological formulation | Instructions are written using formal terminology from algebraic topology, such as simplex, homology, and filtration. |
| Graph-theoretic formulation | Tasks are described using standard graph terminology, avoiding domain-specific topological language. |
| Minimal formulation | Prompts are short and generic, with little or no explicit context, requiring the model to infer the task from minimal cues. |

