# OpenReview forum: "Large Language Models as Topological Thinkers: A Benchmark on Graph Persistent Homology"
_ICML.cc/2026/Conference — ICML 2026 regular_

### Official Review · Reviewer_k3Rg · 2026-02-26

**Soundness:** 4
**Presentation:** 4
**Significance:** 2
**Originality:** 3
**Overall Recommendation:** 4
**Confidence:** 5

**Summary:**

Topological data analysis (TDA) is a fairly new -- or at least fairly newly relevant -- mathematical field that concerns itself with the use of topological tools to better understand high dimensional data. One of the primary cogs of this field is the notion of persistence homology (PH). This paper establishes benchmarks to test how well an LLM can reason in the language of persistence.

**Compliance With Llm Reviewing Policy:**

Affirmed.

**Final Justification:**

During the discussion period I improved my score from a 3 to a 4. I have decided that I will maintain this latter score.

On the positive side, this work considers the intersection of two topics that (to my knowledge) have never spoken to each other in the past. The idea to create benchmarks to analyze the efficacy of LLMs in persistence homology is a genuinely novel one. To use persistence and topological data analysis as somewhat of a proxy to determine how well LLMs reason with topology more broadly is exciting to me. The ultimate results of this paper are both extremely expected (LLMs are still quite poor at topology) and extremely unexpected (certain LLMs are astonishingly better at certain tasks than others). The presentation of the work is absolutely pitch perfect, and perhaps the best literal reading experience that I had as a reviewer for this conference.

On the negative side, While this work's statistical analyses point to a number of exciting possibilities none of them are ultimately explored. While reading I could not get over the impression that this work was the first (relatively small) step in what could become something monumental. Moreover, while working at the intersection of two very unrelated fields has its benefits, one cannot ignore the obvious cons as well. It is possible that this project is ultimately too niche to be truly appreciated. In my initial score, I leaned on the more negative side (3) because I wasn't quite sure whether ICML was the correct venue for such a "first step." However, after reading other reviewers and rebuttals, I ultimately leaned more positive.

**Key Questions For Authors:**

I have no specific questions for the authors. As I stated in my review, the presentation of this work left no room for confusion or concern. I just wanted to take this space to say that I truly did enjoy reading this work. I just wish there was a bit more to your conclusions.

**Limitations:**

Yes

**Strengths And Weaknesses:**

Soundness: The mathematics discussed in this work is perfectly sound. The benchmarks developed by the authors do indeed seem to very thoroughly test the performance of LLMs in reasoning with topological data

Presentation: This paper was by far the best written of all of the papers I was given to review. Every single piece of it is explained with such beautiful concise prose, and at no point did I find myself lost or unsure what the authors were doing. Every step taken clearly follows the previous step, and summaries are provided for every subsection to reinforce learning.

Significance: At this point in time, benchmarks are becoming more and more significant as researchers and even just the general population use LLMs in every aspect of their work and life. This particular benchmark concerns itself with a quite niche aspect of mathematical research. This is both a pro and con of this work. While it is certainly the case that researcher in TDA might find these benchmarks illuminating, I don't see them as having strong significance outside of this very specific research area, beyond the normal way that all research ultimately inspires the next piece of research. In other words, in so far as significance is concerned, this work is at best the first step in a much larger project. This is especially so considering just how many strange behaviors and lingering questions the authors observe

Originality: This work is certainly original in that it deals in two areas that have thus far not really spoken to each other at all. It is notable, however, that many of the conclusions reached are exactly what one would expect. In very broad terms these benchmarks reveal that, outside of the very simplest counting tasks, LLMs are generally quite poor at reasoning in any deeply mathematical way. That being said, however, the devil as always is in the details. It is fascinating and somewhat inexplicable that while most models perform poorly, certain models perform so much better at certain tasks than others. As hinted at in the previous section, all of these comparisons suggest something genuinely deep and strange is going on with the inner workings of the various models that we have no explanation for. Unfortunately, however, any kind of study as to what these deeper implications are, is not part of this work. Once again, this feels like the first step in a much larger research project.

---

> ### Author Rebuttal · Authors · 2026-03-31
>
> Thanks a lot for the careful and generous reading, and we are genuinely glad that the mathematical soundness and presentation met your expectations.
>
> LLM4PH targets two complementary goals: evaluating LLMs as potential assistants in PH-based workflows, and using PH reasoning tasks as a structured probe of LLM capabilities more broadly. Both motivations benefit from a systematic decomposition of the PH pipeline into subtasks, which is what this benchmark provides.
>
> Beyond the main results, we have taken several steps to understand the observed patterns more deeply. Our cross-task analyses include prompt encoding and terminology ablations across multiple graph representations, post-training experiments on LLaMA showing that supervised adaptation on small graphs transfers to larger ones, and a compositional pipeline evaluation on real-world data. These experiments were designed to move beyond aggregate scores and expose where and why models fail.
>
> Our results also surface several non-obvious findings that we believe point toward concrete future directions.
>
> - First, there is a striking performance inversion in the real-world tasks: Filtration Selection, which asks the model to choose one of six candidate functions, scores near-random across all models (0.32-0.46), while Filtration Sequence Generation, which requires constructing a length-5 sequence, reaches 0.95-0.97 for top models on LDHFR. This inversion reveals that LLMs cannot reason declaratively about which filtration strategy will produce discriminative topological features, but can make meaningful local adjustments when the strategy is fixed.
>
> - Second, the failure on Filtration Selection is not random: most models systematically default to selecting edge weight regardless of graph structure, suggesting a heuristic bias toward interpretable choices over topologically informed ones.
>
> - Third, DeepSeek-R1 achieves 96.0% on Simplex Birth Time but collapses to 5.0% on Component Merge Time for large graphs. Both tasks track a topological event along a filtration sequence, yet the first requires only a localized scan while the second requires maintaining a global connectivity state that grows with graph size. This dissociation surfaces a distinction between sequential lookup and dynamic state tracking that simpler graph benchmarks do not separate.
>
> We see these patterns as a natural starting point rather than a complete account, and we hope this work provides a useful foundation for the larger research project you envision. We would appreciate your reconsideration of the significance of this contribution.

---

> > ### Author Rebuttal · Reviewer_k3Rg · 2026-03-31
> >
> > My biggest concern while reading this work was that there was a sense that it was the beginning of something potentially major, but not necessarily major on its own terms. I do still feel this way, to a certain extent. My grade therefore comes down to the question of whether ICML is the right venue for work that points to a number of possibly interesting deeper phenomena, but does not really engage with trying to understand the source of any of them. The author's rebuttal was helpful in allowing me to consolidate a number of thoughts that I was having about the specific technical achievements of the paper.
> >
> > After thinking about this for a little while, I have ultimately decided to lean slightly more positively, and change my score from a 3 to a 4.

---

> > > ### Author Response · Authors · 2026-04-01
> > >
> > > Dear Reviewer k3Rg,
> > >
> > > Thanks so much for your feedback, for all the helpful comments, thought provoking questions, recognizing the potential of this paper to start something major and drastically new,  and, of course, for raising the score!
> > >
> > >  Toward your point on whether ICML is the right venue. We share some of your thoughts on it, but we believe that no matter what, such fundamental interplay between LLM, PH, and graph learning will eventually reach the ML community. The earlier it happens, the more edge the ML community will have to understand the underlying phenomena, all gaps and ways to address them.
> > >
> > > Following your suggestions,  we'll incorporate your questions in a form of conjectures/open problems into the camera ready version, trying to pave a way for some bigger and deeper research on these topics.
> > >
> > > Sincerely,
> > > the Authors.

---

### Official Review · Reviewer_3nBH · 2026-03-05

**Soundness:** 3
**Presentation:** 3
**Significance:** 3
**Originality:** 2
**Overall Recommendation:** 5
**Confidence:** 4

**Summary:**

The ability of LLMs to understand and reason over graphical structures has been widely debated and investigated in recent years. This research direction is primarily motivated by the inherent encoding capabilities of graphical structures, which, in principle, could be combined with the reasoning abilities of LLMs to produce valuable outcomes across numerous applications. Based on that view, this work proposes a formal, elaborate, and diverse benchmarks for better assess the graph-related capabilities of LLMs. The authors argue that the appropriate language of such a benchmark should be that of Persistent Homology (PH) and they propose a benchmark of about 10K samples, that span across 10 different questions of various difficulties.

**Compliance With Llm Reviewing Policy:**

Affirmed.

**Final Justification:**

The score fully describes the paper: Technically solid paper, with high impact on at least one sub-area of AI. Given the authors' responses, I believe this is the most appropriate score.

**Key Questions For Authors:**

Existing works have shown that (a) the conversion of a graph to a textual description is non-trivial and different approaches yield different results, and (b) LLMs are not so capable of understanding textual descriptions of graphical structures, especially as their size and complexity grows. Have you experimented with any of the existing Graph Language Models that employ a graph encoder in conjunction with an LLM?

**Limitations:**

yes

**Strengths And Weaknesses:**

### Strengths

1. The paper is quite clear and well-written. The main idea is properly conveyed and the experiments are well-studied and presented.

2. There is indeed a need for formal and diverse graph+language benchmarks, and this paper addresses this. Multiple works in that area rely on relatively small public datasets or, even worse, on custom-made datasets that are not used across all works.

3. The corresponding benchmark covers multiple different aspects of graph questions, from basic ones (e.g., counting), to more advanced ones (e.g., filtration generation).

### Weaknesses

1. The main weakness of this paper is the fixation with PH terms. It is mentioned quite some times that the bad results of the LLMs show that they can not really understand graphical structures in depth (e.g., lines 233, 258, and 302). However, as it is well known, the quality of the prompt as well as its specific phrasing play a crucial role in the performance and the response of the LLM. It is not clear from the experiments whether the LLMs do not understand the graphical structures or, simply, they are not that familiar with the PH terms that the prompts ask about.
For instance, in the task 1D simple counting, it would be very important to present the performance of LLMs when they are prompted to answer the same question but without mentioning any of the, perhaps confusing, PH terms. Then, the paper's conclusion would be better grounded to the experimental evidence.

2. In general, I think there should be some ablation studies that demonstrate the effect that different wording of the concepts have on the results. Perhaps PH is the language to go (as the authors mention), but this does not mean that current LLMs are capable of understanding it, just like older versions of LLMs couldn't understand languages beyond English. Knowing whether the issue lies in the PH language or whether it is a more general inability of the LLMs to understand graphs is an important and invaluable insight.

3. The lack of any prompt examples makes it very difficult to better understand the questions and the degree of difficulty they pose. In Table's 12 caption it is mentioned _"Full prompt examples are provided in the appendix."_, however no such examples are provided.

---

> ### Author Rebuttal · Authors · 2026-03-31
>
> Thank you very much for the thoughtful review!
>
> **1. PH terms vs. genuine reasoning gap**
> Terminology does matter, and we take this concern seriously. Table 8 directly addresses it by comparing topological, graph-theoretic, and minimal formulations across multiple tasks and models. The results show that sensitivity to phrasing is real but task-dependent. For example, GPT-4o on Component Count Under Filtration scores 0.633 under topological phrasing and 0.785 under graph-theoretic phrasing. However, on Component Merge Time, the same substitution produces no consistent benefit: GPT-4o scores 0.510 under topological phrasing and 0.425 under graph-theoretic phrasing, and both models remain far from ceiling regardless of formulation.
>
> This suggests that terminology explains part of the difficulty on some tasks but not the failures on filtration-evolution tasks more broadly. Table 12 summarizes the full set of prompt styles used in our ablations. The actual prompt examples are currently missing from the appendix, which makes it harder to evaluate this distinction directly. We will include complete prompt examples for all formulation types in the revision.
>
> **2. Wording ablations are part of the design**
>
> Table 8 provides evidence that both failure modes exist. On Component Count Under Filtration, switching to graph-theoretic phrasing improves performance, suggesting that PH terminology creates a genuine barrier for some tasks. On Component Merge Time, the same switch produces no meaningful improvement: the task asks simply when two nodes first become connected as edges are added one by one, and models still fail. The distinction between tasks where language is the obstacle and tasks where the reasoning itself breaks down is precisely the insight the reviewer is asking for. We will make this analysis explicit in the revised discussion and include full prompt examples in the appendix.
>
> **3. Prompt examples**
>
> We apologize for this oversight. The prompt examples were inadvertently omitted from the submitted appendix despite being referenced in Table 12. We will include them in the revision and correct the caption accordingly.
>
> **4. Graph textualization and graph-encoder models**
>
> Graph representation is treated as an explicit variable in our benchmark rather than a hidden confound. Table 8 compares text-style, code-like, and matrix-based encodings across tasks, and Table 7 examines larger graphs via ego-Facebook subgraph sampling to study how performance degrades with scale.
>
> LLM4PH evaluates PH-style reasoning in a text-based interface by design. In this setting, graph textualization is part of what is being tested: the benchmark asks whether an LLM can reason about filtration-based structural evolution when graph information is presented through language. A graph-encoder + LLM model would address a different question, namely whether a stronger graph interface reduces the observed gap. We consider this complementary and will make this scope boundary explicit in the revision.

---

> > ### Author Rebuttal · Reviewer_3nBH · 2026-04-01
> >
> > I thank the authors for their response. I will raise my score. Besides the current work, I suggest a work on graph encoder + LLM could be a great follow up which could provide better results and more informative feedback to the community.

---

> > > ### Author Response · Authors · 2026-04-02
> > >
> > > Dear Reviewer 3nBH,
> > >
> > > Thanks a lot for your suggestion of graph encoder + LLM (we will certainly involve the discussion and suggested works in the final version)! Thanks very much again for your effort in reviewing our paper and re-evaluating it. We are very grateful for the constructive and inspiring feedback and of course for raising the score!
> > >
> > >
> > > Best,
> > >
> > > Authors

---

### Official Review · Reviewer_Y3mn · 2026-03-10

**Soundness:** 3
**Presentation:** 3
**Significance:** 2
**Originality:** 2
**Overall Recommendation:** 4
**Confidence:** 3

**Summary:**

The work evaluates whether large language models (LLMs) can reason about multi-scale structural evolutions in graph from the perspective of persistent homology (PH). Compared with existing benchmarks, this paper employs PH to test higher-order and cross-scale reasoning ability of LLMs. Concretely, LLM4PH incorporates 4 task categories and 10 subtasks, ranging from simplicial structure understanding, filtration evolution reasoning, filtration strategy design to real-world graph inference. The experiments compare a range of private and open-source LLMs, and the main finding is that models may do reasonably well on local tasks, but generally struggle on dynamic structural evolution.

**Compliance With Llm Reviewing Policy:**

Affirmed.

**Final Justification:**

I think this benchmark can evaluate the some part of the LLMs' ability, and the rebuttal has addressed my concerns. For this work, I think it is a good benchmark for understanding whether LLM understands topology.

**Key Questions For Authors:**

1. Why do you choose these datasets, BZR, COX2 and LDHFR, as real-world benchmarks?

**Limitations:**

yes

**Strengths And Weaknesses:**

Strengths:

1. The motivation is clear and interesting. The paper identifies a real gap in current graph reasoning benchmarks, whereas this benchmark targets global structural evolution across scales.

2. Benchmark design is broad and reasonably well structured. LLM4PH is designed as a progressive pipeline as simple tasks (Simplicial Structure Understanding), medium tasks (Filtration Evolution Reasoning), hard tasks (Filtration Strategy Design).

3. The experiments is quite extensive. The paper evaluates multiple LLMs and provide abundant analysis on cross tasks.

Weaknesses

1. Although the authors extend the current benchmark to a more challenging task, I think the paper will be much solid if the work can derive any conclusion that contradicts the results derived from prior works. Currently, it seems this is just another benchmark, but doe not bring very interesting insights towards our understanding to LLMs.

2. It is still challenging for the benchmark to provide a totally fair prompting method. The paper per se also finds the prompting makes a great difference, but it also weakens the claim that the benchmark cleanly isolates reasoning ability rather than prompt sensitivity.

3. The real-world evaluation is somewhat not strong enough. The real-world tasks use small graph sets from three molecular datasets, and the direct-classification baseline is almost perfect. This raises the concern that the PH-guided tasks may be artificially difficult, rather than revealing a clean reasoning gap. In other words, the downstream task setup may not convincingly demonstrate the value of PH as an evaluation metric.

4. There is no solid strong comparison to specialized baselines. It would strengthen the paper to compare against non-LLM baselines. If some simples baselines can address these problems well, it seems it would be unnecessary to involve the LLMs into such problems. The paper mostly concludes that pure LLMs struggle, which is interesting but somewhat unsurprising.

---

> ### Author Rebuttal · Authors · 2026-03-31
>
> Thank you very much for the thoughtful review!
>
> **1. Limited insight**
>
> LLM4PH does reveal something that prior graph benchmarks cannot: a precise dissociation in how LLMs handle topological reasoning. Models are not uniformly weak on PH tasks. They fail systematically on abstract judgment tasks while succeeding on concrete adjustment tasks, and this boundary is consistent across models and graph sizes.
> The clearest evidence is the performance inversion in Table 5. From a task-design standpoint, Filtration Selection, which asks the model to choose one of six candidate functions, should be easier than Filtration Sequence Generation, which asks the model to construct a length-5 sequence from ten possible values. Yet all models score near-random (0.32-0.46) on Filtration Selection while top models reach 0.95-0.97 on Filtration Sequence Generation for LDHFR. This is not a marginal difference but a qualitative inversion.
>
> We think this reflects a specific property of LLM reasoning that goes beyond PH: when a task requires declarative judgment about which abstract strategy will produce a desired global outcome, LLMs fall back on surface heuristics. Our output analysis shows that most models systematically select edge weight as the filtration function regardless of graph structure, which is the most interpretable choice but not necessarily the most discriminative one. When the same models are instead asked to adjust concrete parameters under a fixed strategy, they perform meaningfully. Prior graph benchmarks do not separate these two modes of reasoning. LLM4PH does, and we believe this dissociation is informative beyond the topological setting. We will foreground this analysis in the revised discussion.
>
> **2. Prompt fairness**
>
> Prompt sensitivity is a real phenomenon in LLM evaluation, and we treat it as a variable to be measured rather than a confound to be eliminated. Our prompt-encoding and terminology ablations quantify exactly how much wording and graph representation affect performance across different task types. This gives a more complete picture of model capability than a single fixed prompt would provide.
>
> Importantly, the ablations also show that prompt sensitivity does not fully explain the observed difficulty. Even under the most favorable prompt conditions, filtration-evolution and filtration-design tasks remain substantially harder than simpler structural tasks. The reasoning gap is therefore real and not an artifact of prompt choice. We will clarify this framing more explicitly in the final version.
>
> **3. Real-world evaluation**
>
> The near-perfect direct-classification results do not contradict the difficulty of the PH-guided tasks because the two settings test fundamentally different things. Direct classification asks whether a model can distinguish graph classes using any available signal. The PH tasks ask whether a model can reason about which filtration strategy will expose topologically meaningful differences between classes. A model can succeed at the former through shallow pattern matching while failing at the latter.
> The contrast between the two PH tasks also argues against the setup being artificially difficult. Filtration selection scores near-random across all models, while filtration-sequence generation reaches 0.95-0.97 for top models on LDHFR. If the tasks were uniformly arbitrary, we would not expect this kind of structured variation. We will make this framing more explicit in the revision.
>
> **4. Specialized baselines**
>
> Table 6 already includes non-LLM baselines in the form of pure code-based PH pipelines. Betweenness centrality reaches 0.890 and Closeness reaches 0.800 on BZR, showing that when the right filtration is given, the downstream computation is well-handled by existing tools. The open question is who decides which filtration to use. This decision currently requires domain expertise: a researcher applying PH to a new dataset must manually select the filtration function based on domain knowledge. LLM4PH is precisely about whether LLMs can take on this role, reasoning from graph structure and task objectives to appropriate filtration strategies without human intervention.
>
> **5. Why BZR / COX2 / LDHFR?**
>
> These three molecular graph datasets are standard benchmarks in the TDA and graph classification literature. BZR and COX2 in particular have been used in prior PH-based graph learning work[1]. Beyond this, we chose them for three practical reasons: their graph sizes are small enough to be represented as text within LLM context windows, their class labels are clean and unambiguous, and their classification setting maps naturally onto our filtration selection and filtration sequence generation tasks. For scalability beyond the molecular setting, Table 7 evaluates models on ego-Facebook subgraphs, where graph size is substantially larger and the structural complexity is qualitatively different from molecular graphs.
>
> [1] Hofer C, Graf F, Rieck B, et al. Graph filtration learning

---

> > ### Author Rebuttal · Reviewer_Y3mn · 2026-04-02
> >
> > Thanks for the rebuttal, which has addressed my concerns. I will accordingly raise my score.

---

> > > ### Author Response · Authors · 2026-04-02
> > >
> > > Dear Reviewer Y3mn,
> > >
> > > Thank you very much for your positive feedback and for acknowledging our rebuttal.
> > >
> > > We are glad that our responses have addressed your concerns, and we sincerely appreciate your careful evaluation and consideration. Your comments have helped us further improve the clarity and presentation of the paper.
> > >
> > > Best,
> > >
> > > Authors

---

### Official Review · Reviewer_M6a7 · 2026-03-12

**Soundness:** 3
**Presentation:** 2
**Significance:** 3
**Originality:** 3
**Overall Recommendation:** 4
**Confidence:** 2

**Summary:**

This paper proposes LLM4PH, a benchmark for evaluating the capabilities of LLMs on persistent homology of graphs. The authors first point out that persistent homology (PH) provides a good view of global structures of graphs. Then they propose three kinds of tasks: simple ones (Simplicial Structure Understanding), medium ones (Filtration Evolution Reasoning), and hard ones (Filtration Strategy Design). Their real-world tasks and extensive experiments provide a comprehensive view of the critical problem.

**Compliance With Llm Reviewing Policy:**

Affirmed.

**Final Justification:**

I recommend 4 for this paper

**Key Questions For Authors:**

I discover that in Table 4 (Non-Uniform Filtration Generation), the standard deviations are very large. The authors explain that unstable outputs can be seen, but I want to see some examples to know on what tasks LLMs perform well, and on what tasks not so well.

Additionally, you can answer my questions about motivation and dataset construction.

Due to the missing explanations, I strongly suggest that the authors modify the paper carefully and provide a better version (either in the final version or the next submission)

**Limitations:**

They do not include the limitations and potential negative societal impact of their work. However, this is a benchmark mainly for an aspect of capabilities on understanding graphs of LLMs, there might not be any potential negative societal impacts.

**Strengths And Weaknesses:**

Strengths

++ The task categorization is clear. There are categories of problem difficulty (simple, medium, hard) provided, and also sub-tasks under each category. This will provide a hierarchical view of the LLMs' capabilities. Additionally, the proposed graph tasks are professional. Unlike other works on LLM+graph, this paper dives into the research field of inner mechanisms on LLMs for graph tasks.

++ The real-world graph inference tasks help me to understand why PH is a crucial thing in LLMs for graphs.

++ Evaluated LLMs are recent LLMs, and this keeps the evaluation aligned with the latest research advances. On each task, the authors provide an observation, clearly demonstrating the findings.

Weaknesses:

-- In Page 1, you can use some toy examples to demonstrate your motivations and key points. I have known that PH is important in traditional graph structure learning (from your references, text explanations), but a figure of toy examples and explanations will provide obvious understandings. Additionally, why PH is also important for LLMs to understand a graph? Besides 'a lens for probing', maybe you can clearly explain the relationship between PH and LLMs for graphs with cases. For example, will an LLM perform well on downstream tasks of graphs if it performs well on LLM4PH?

-- I cannot find the detailed data construction process in the main text. Appendix B shows the process, but I guess it may be better to move it to the main text.  The data quality is very important in benchmark construction, and I'm somewhat confused on how the authors ensure the quality of the benchmark and the ground truth, like human/GPT verification, data cleaning, etc. I appreciate the authors' efforts on graphs, however, since this is a benchmark for evaluating LLMs, more details of the 'language part' should be included. Due to this fact, the 'benchmark design' section is somewhat short for detailed explanations of technical contribution.

-- Some definitions of the tasks are not so easy to understand. The authors provide brief descriptions of each sub-task, however, there should be some explanations on concepts like K-shell, closeness centrality, betweenness centrality, or eigenvector centrality (I have read other papers and blogs to understand them). There should also be some extra visualizations & comparisons for the tasks. This suggestion is provided because the task definitions are highly correlated to the task difficulty.

-- The applied LLMs in the experiments are mainly general LLMs. I'm curious about how will the task-specific LLMs ( here I do not mean LLMs fine-tuned on PH datasets, but some LLMs that can integrate graph information or fine-tuned on other graph-based datasets, like [1][2] and others. ) perform, on your proposed LLM4PH. Additionally, the utilized models are not cited (Experimental Setup).

-- The authors should also provide some real model responses either in the main text or the Appendix. These qualitative experiments will help readers better understand the models' capabilities.

[1] Wang, Jianing, et al. "Instructgraph: Boosting large language models via graph-centric instruction tuning and preference alignment." Findings of the Association for Computational Linguistics: ACL 2024. 2024.
[2] Chen, Nuo, et al. "Graphwiz: An instruction-following language model for graph computational problems." Proceedings of the 30th ACM SIGKDD Conference on Knowledge Discovery and Data Mining. 2024.

---

> ### Author Rebuttal · Authors · 2026-03-31
>
> We thank the reviewer a lot for the careful reading and constructive suggestions.
>
> **1. Toy example / motivation**
> We will add a toy filtration example on Page 1 showing two graphs that appear identical under static analysis but produce different persistence barcodes because their edges are added in different orders under a weight-based filtration.
> The PH pipeline involves a sequence of distinct reasoning steps including structural recognition, filtration evolution tracking, and filtration strategy design. LLM4PH asks where in this pipeline LLMs can contribute reliably and where they cannot. Our compositional pipeline results in Table 6 show that LLM capability at specific pipeline steps has measurable consequences for end-to-end performance, making LLM4PH a diagnostic tool for mapping LLM reasoning capacity onto the specific demands of the PH workflow.
>
> **2. Data construction**
> Our ground truth is **algorithmically generated from PH computations**, not by human/GPT annotation, which ensures reproducibility and eliminates annotation noise. We will expand the Benchmark Design section to make this pipeline more explicit, including quality control, balanced sampling, and the language-side design (representative prompts and graph encodings).
>
> **3. Task definitions**
> These centrality measures serve as candidate filtration functions, and their structural interpretations directly determine what topological patterns each filtration reveals. For example, node degree builds the graph from hubs outward, capturing community-level connectivity. K-shell reflects hierarchical core-periphery structure through iterative pruning.
>
> Because each function reveals different structural properties, the resulting persistence barcodes can differ substantially on the same graph, which is what makes filtration selection a nontrivial reasoning task. We will add a comparison figure and concise definitions in the revised paper.
>
> **4. Graph-specialized LLM baseline**
>
> We added a graph-specialized baseline starting from the GraphWiz LLaMA2-7B-DPO checkpoint, further adapted with LoRA using 700 task-format dialogue instances. GraphWiz produced format-compliant outputs on Tasks 1 and 2 only; Tasks 3 and 4 suffered from format non-compliance and are reported for completeness.
>
> | Task | GraphWiz | Best in paper | Output validity |
> | ----- | ----- | ----- | ----- |
> | 0D Component Counting (S) | 0.110 | 1.000 | Format-compliant |
> | Component Merge Time (S) | 0.067 | 0.470 | Format-compliant |
> | Non-Uniform Filtration Generation (S, rank↓) | 127.0 | 43.4 | Partial format failure |
> | Direct Classification (BZR) | 0.000 | 0.995 | Format failure |
>
> On the two format-compliant tasks, GraphWiz scores 0.110 on Component Counting and 0.067 on Component Merge Time, far below the strongest general LLMs (1.000 and 0.470 respectively). This performance gap is not explained by instruction following, since outputs are format-compliant on these two tasks. It suggests that graph-task tuning on standard graph benchmarks does not equip a model with the filtration-based reasoning that LLM4PH requires. The capability probed here, tracking topological feature evolution across a filtration sequence, appears distinct from what graph-specialized training currently provides. We will include these results and the missing model citations in the revision.
>
> **5. Real model responses**
> Yes, qualitative responses are important for grounding the aggregate metrics. We will add representative successful and failed outputs in the appendix, especially for Component Merge Time and Non-Uniform Filtration Generation, to make the observed failure modes and capability differences more concrete.
>
> **6. Table 4 variance**
> Models perform well when there is a clear local anchor in the task. On Component Count Under Filtration, the model only needs to determine how many components exist at a specific threshold, and DS-R1 reaches 97.5% even on large graphs. On Non-Uniform Filtration Generation, the model must choose a sequence of five thresholds such that the resulting barcodes of two graphs are as different as possible. There is no single step that can be verified along the way, and the effect of each threshold choice depends on all the others. Models tend to produce very different outputs across runs without converging on a good solution, which explains why the standard deviations in Table 4 are nearly as large as the mean ranks. We will add representative output examples in the revision.

---

> > ### Author Rebuttal · Reviewer_M6a7 · 2026-04-01
> >
> > I have read the rebuttal, and thanks for the detailed response.
> >
> > Questions (sorry for adding an "official comment" that cannot be seen by you yesterday):
> >
> > 1. Thanks for this clarification on toy examples. I guess that you can represent it by texts here? Like this (nodes+links) :1-->2, 7-->4, ...
> >
> > 2. Data construction: I appreciate the clarification on data construction. I think that the discussion of this part is important.
> >
> > 3. Are there any formal standards for defining the difficulties of tasks? Sorry for not very familiar with this area.
> >
> > 4. Could you provide some analysis on, why the graph-based LLMs fail in your tasks?
> >
> > 5. I now understand why the models show a very large standard deviation.
> >
> > I will update my score since this is an interesting work, and this work should be fairly evaluated though I do not have much experience in this area.

---

> > > ### Author Response · Authors · 2026-04-01
> > >
> > > Thanks very much for the response, Reviewer M6a7! We'll be waiting for your follow up questions.
> > >
> > > The authors.

---

### Decision · Program_Chairs · 2026-04-30

**Decision:**

Accept (regular)

**Comment:**

The authors propose LLM4PH, a benchmark for evaluating LLMs on persistent-homology-based reasoning over graphs. All reviewers noted that the benchmark is well-motivated, well-structured, and technically meaningful. In particular, the task hierarchy, the use of persistent homology to study global graph structure, and the breadth of the evaluation were viewed as strengths.

Some reviewers asked for clearer motivation for why PH is the right language for probing LLM graph reasoning, more details on benchmark construction and task difficulty, and more specialized graph-oriented baselines. However, these concerns were successfully addressed in the rebuttal through clarifications and the addition of baseline results.

All reviewers recommend accepting the paper. The paper would benefit from a clearer presentation of the benchmark. However, the core contribution appears useful, novel, and likely to support future work at the intersection of LLMs, graphs, and topology. I would lean toward acceptance.